# Quantification of human papillomavirus cell-free DNA from low-volume blood plasma samples by digital PCR

Fabian Rosing,[1] Matthias Meier,[2] Lea Schroeder,[1] Simon Laban,[2] Thomas Hoffmann,[2] Andreas Kaufmann,[3] Oliver Siefer,[4] Nora Wuerdemann,[5] Jens Peter Klußmann,[4] Thorsten Rieckmann,[6,7] Yvonne Alt,[1] Daniel L. Faden,[8,9] Tim Waterboer,[1] Daniela Höfler[1]

**ABSTRACT** The incidence rate of human papillomavirus-driven oropharyngeal cancer (HPV-OPC) is increasing in countries with high human development index. HPV cell-free DNA (cfDNA) isolated from 3 to 4 mL blood plasma has been successfully used for therapy surveillance. A highly discussed application of HPV-cfDNA is early detection of HPV-OPC. This requires sensitive and specific cfDNA detection as cfDNA levels can be very low. To study the predictive power of pre-diagnostic HPV-cfDNA, archived samples from epidemiological cohorts with limited plasma volume are an important source. To establish a cfDNA detection workflow for low plasma volumes, we compared cfDNA purification methods [MagNA Pure 96 (MP96) and QIAamp ccfDNA/RNA] and digital PCR systems (Biorad QX200 and QIAGEN QIAcuity One). Final assay validation included 65 low-volume plasma samples from oropharyngeal cancer (OPC) patients with defined HPV status stored for 2–9 years. MP96 yielded a 28% higher cfDNA isolation efficiency in comparison to QIAamp. Both digital PCR systems showed comparable analytical sensitivity (6–17 copies for HPV16 and HPV33), but QIAcuity detected both types in the same assay. In the validation set, the assay had 80% sensitivity ($n = 28/35$) for HPV16 and HPV33 and a specificity of 97% ($n = 29/30$). In samples with ≥750 µL plasma, the sensitivity was 85% ($n = 17/20$), while in samples with <750 µL plasma, it was 73% ($n = 11/15$). Despite the expected drop in sensitivity with decreased plasma volume, the assay is sensitive and highly specific even in low-volume samples and thus suited for studies exploring HPV-cfDNA as an early HPV-OPC detection marker in low-volume archival material.

**IMPORTANCE** HPV-OPC has a favorable prognosis compared to HPV-negative OPC. However, the majority of tumors is diagnosed after regional spread, thus making intensive treatment necessary. This can cause lasting morbidity with a large impact on quality of life. One potential method to decrease treatment-related morbidity is early detection of the cancer. HPV cfDNA has been successfully used for therapy surveillance and has also been detected in pre-diagnostic samples of HPV-OPC patients. These pre-diagnostic samples are only commonly available from biobanks, which usually only have small volumes of blood plasma available. Hence, we have developed a workflow optimized for small-volume archival samples. With this method, a high sensitivity can be achieved despite sample limitations, making it suitable to conduct further large-scale biobank studies of HPV-cfDNA as a prognostic biomarker for HPV-OPC.

**KEYWORDS** HPV, liquid biopsy, digital PCR, OPC, cfDNA, early detection

Infection with high-risk human papillomavirus (HPV) is one of the leading causes of cancer worldwide, accounting for 5% of all cases (1). HPV is a sexually transmitted infection and is primarily associated not only with cancers of ano-genital sites, such as vulvar, penile, and anal carcinomas, and particularly with more than 99% of all cervical

Address correspondence to Daniela Höfler, d.hoefler@dkfz.de.

T.W. serves on advisory boards for Merck, Sharp & Dohme (MSD). S.L. serves on advisory Boards of MSD, Bristol Myers, Squibb (BMS), and Astra Zeneca (AZ) and has received honoraria from MSD, BMS, AZ, and Merck Serono. D.L.F. has received research funding or in-kind funding from BMS, Calico, Predicine, BostonGene, and NeoGenomics and has received consulting fees from Merck, Noetic, Chrysalis Biomedical Advisors, Arcadia, and Focus. He receives salary support from National Institutes of Health (NIH)/National Institute of Dental and Craniofacial Research (NIDCR) K23 DE029811, NIH/NIDCR R03DE030550, and NIH/National Cancer Institute R21CA267152.

See the funding table on p. 14.

carcinoma cases (1) but also with head and neck cancer, such as oropharyngeal cancer (OPC) (2). Due to a strong increase of HPV-driven OPC, OPC is now the most frequent HPV-driven cancer in some countries with high human development index, especially in North America and Western Europe (2). HPV-driven OPC mostly arises from the epithelium of the tonsillar crypts and the base of the tongue and is often diagnosed after regional spread to nearby lymph nodes, making extensive treatment necessary (3). The clinical standard for determining the HPV status of OPC is p16-immunohisto-chemistry (p16-IHC) (4), often accompanied by HPV DNA or RNA detection methods. Despite the overall good prognosis of HPV-driven OPC, survivors often experience significant morbidity after treatment, particularly dysphagia (5). Early detection of OPC may decrease the need for extensive treatment, possibly leading to an improved quality of life. This is complicated by the fact that, contrary to cervical carcinoma, no pre-can-cerous lesions of OPC are known to date (6). Therefore, screening and early detection (i.e., secondary prevention) are currently not feasible. Primary prevention through HPV vaccination is highly effective at preventing cervical cancer and, expectedly, HPV-driven OPC as well, as the vaccine reduces the prevalence of oral high-risk HPV infection (7). However, due to OPC mainly affecting males and the low vaccine uptake in males, HPV vaccination is not expected to have an impact on human papillomavirus-driven oropharyngeal cancer (HPV-OPC) incidence rate for decades to come. Hence, HPV-driven OPC incidence rates are projected to keep increasing until at least the mid-2030s (8).

HPV cell-free DNA (cfDNA) from blood plasma has recently gained significant attention. Cells that undergo apoptosis or necrosis shed DNA into their environment and subsequently also the bloodstream (9). In healthy individuals, plasma-derived cfDNA is mostly derived from hematopoietic cells (10), but in cancer patients, a significant fraction can be derived from neoplastic cells. In cancer patients, the proportion of tumor-derived cfDNA varies from less than 5% to more than 90% (9). Overall, cfDNA concentrations can vary widely between patients, with typical values spanning from 10 to 1,200 ng/mL (9). As cfDNA is often of very low abundance and highly fragmented, with a typical fragment size of ~165 bp (9), specialized protocols and assays are necessary for its analysis. Detection of HPV cfDNA in blood plasma can identify OPC patients with high sensitivity and specificity (11) and has been successfully used for diagnosis, prediction of recurrence, and monitoring therapy response in HPV-driven OPC and other HPV-driven head and neck cancers (12–16). HPV cfDNA presence after intended curative treatment is a highly specific predictor of recurrence, and recurrence has been detected up to 19 months earlier by HPV cfDNA in comparison to standard imaging techniques (17). Therefore, HPV cfDNA is a promising candidate biomarker for a screening approach due to being minimally invasive and being a highly sensitive and specific biomarker for HPV-driven malignancies (18). Currently, the standard method for analyzing cfDNA is digital droplet PCR (ddPCR). ddPCR relies on splitting the PCR into many thousand partitions, usually by generating droplets in an emulsion (e.g., BioRad QX200), but nanoplate-based approaches exist as well (e.g., QIAGEN QIAcuity), then referred to as digital PCR (dPCR). The market for dPCR devices is rapidly evolving, with new devices being capable of detecting multiple targets at once. While the vast majority of HPV-OPCs are caused by HPV16, about 5%–10% of cases are caused by other genotypes, especially in relatively high-incidence countries (14). In plasma samples at OPC diagnosis, the sensitivity of ddPCR has been reported to range from 70% (19) to more than 98% (14). ddPCR is more sensitive than qPCR in HPV16-cfDNA detection from OPC and comes with lower costs than sequencing-based approaches, even though the latter have been reported to reach even higher sensitivity than ddPCR (19). Due to the fragmented nature of cfDNA in blood plasma, it is of high importance for any PCR-based method to use very small amplicon sizes to maximize sensitivity of the assay. Most studies have focused on application of the method for therapy surveillance or diagnosis, where 3 mL of plasma or more are relatively easily available for analysis (13, 14, 20). Previously published ddPCR assays targeted the E6 or E7 regions of the HPV genome due to the high sequence conservation of these regions and frequent amplification of these genes, which reduce

the risk of false-negative results (13, 14, 20). In HPV-OPC, deletions of large parts of the HPV genome can occur (21), but as E6 and E7 are important drivers of HPV-associated cancer growth (22), these genes are unlikely to be deleted in HPV-OPC cells. To study the performance of HPV cfDNA as a predictive biomarker of HPV-OPC, it is necessary to analyze existing samples from biobanks or epidemiological cohorts, where often only small amounts of plasma are available.

In this setting, antibodies against HPV16 oncoproteins, especially E6, have been shown to be highly specific and sensitive predictive biomarkers of OPC (23). Seroconversion has been observed to occur up to 10 or more years before diagnosis (24). HPV16 oncoprotein antibody patterns have been successfully used to identify individuals at risk of development of HPV-OPC in a prospective manner before onset of symptoms (25, 26). However, due to the long lead time, repeated in-depth exams by otorhinolaryngologists are necessary over time for early detection of OPC, putting a considerable burden on both the patients and the healthcare system. Therefore, a minimally invasive biomarker for HPV-OPC is necessary in order to correctly identify people at imminent risk of developing HPV-OPC.

The aim of this study was to establish an optimized protocol for the purification and quantification of HPV-cfDNA from small volumes of blood plasma (i.e., below 1 mL). While nucleic acid purification efficiency is usually not a key determinant of cfDNA detection when routine amounts of blood plasma (typically 3–4 mL) are available, it becomes more important when only limited amounts of plasma are available. Blood plasma samples from volunteers were used to compare cfDNA purification methods. On two different digital PCR platforms (BioRad QX200 and QIAcuity One), an assay targeting HPV16, HPV33, and the human reference gene beta-globin (BG) was developed to detect HPV-cfDNA. After determining analytical sensitivity, DNA from cervical swab samples was analyzed to ensure HPV type specificity of the assay. The 89 available liquid biopsy samples from OPC patients were split into a training set of 24 samples and a validation set of 65 samples. The training set was used to optimize the dPCR assay, and final sensitivity and specificity were determined in the validation set.

## MATERIALS AND METHODS

### OPC patient inclusion and testing

Sodium citrate plasma samples from 86 OPC patients from Ulm University Hospital were collected (Table 1). Tumor HPV status was determined by p16-IHC (4) and HPV DNA detection by GP5+/GP6+ PCR (27) and subsequent sequencing of PCR products for HPV typing. OPCs were classified as HPV driven by a pathologist if the tumor section was both HPV-DNA and p16-IHC positive. Tumors which tested negative for either p16 or HPV-DNA (or both) were considered to be HPV negative. Three patients in the validation cohort had one blood sample taken at initial diagnosis and one at recurrence with distant metastasis. Inclusion criteria for the study were the diagnosis of a tumor in the oropharynx with complete tumor HPV-DNA and p16-IHC data. Samples were assigned to the training and validation sets by the treating physician who was not directly involved in the experiments to achieve balanced distributions of HPV-driven and HPV-negative tumors in both sets. The two sets did not differ significantly when examining tumor stage, nodal stage, and other characteristics (Table 1). dPCR experiments were conducted by personnel blinded to tumor HPV status. Available plasma volumes from 24 OPC patients (12 HPV16 driven, 2 HPV33 driven, and 10 HPV negative) in the training set ranged from 0.38 to 2.9 mL (mean volume 1.14 mL). The samples were collected between 2013 and 2020 (mean storage time since blood draw 5.5 years). In the validation set of 65 plasma samples from 62 HPV-OPC patients (33 HPV16 driven, 2 HPV58 driven, 1 HPV33 driven, and 26 HPV negative) available plasma volumes varied between 0.37 and 3 mL (mean volume 1.01 mL).

**TABLE 1** Patient characteristics of the training and validation sets

| Characteristic | Training set (n = 24) | Validation set (n = 62)[a] | P value |
|---|---|---|---|
| HPV type (DNA), n (%) | | | 0.38[b] |
| 16 | 12 (5) | 33 (53) | |
| 33 | 2 (8.3) | 1 (1.6) | |
| 58 | 0 (0) | 2 (3.2) | |
| HPV negative[c] | 10 (42) | 26 (42) | |
| p16-IHC, n (%) | | | 0.89[b] |
| 1 | 14 (58) | 39 (63) | |
| 0 | 10 (42) | 23 (37) | |
| Age at diagnosis (years)[d] | 58 (54–63) | 61 (56, 67)[e] | 0.13[f] |
| Sex, n (%) | | | 0.24[b] |
| Male | 21 (88) | 45 (73) | |
| Female | 3 (13) | 17 (27) | |
| OPC subsite, n (%) | | | 0.24[b] |
| Tonsil | 14 (58) | 48 (77) | |
| Base of tongue | 9 (38) | 11 (18) | |
| Soft palate | 1 (4.2) | 2 (3.2) | |
| Lateral pharyngeal wall | 0 (0) | 1 (1.6) | |
| T stage, n (%) | | | 0.81[b] |
| T1/2 | 13 (54) | 32 (52) | |
| T3/4 | 11 (46) | 29 (47) | |
| Unknown | 0 (0) | 1 (1.6) | |
| N stage, n (%) | | | 0.56[b] |
| N0/1 | 15 (63) | 44 (71) | |
| N2/3 | 9 (38) | 17 (27) | |
| Unknown | 0 (0) | 1 (1.6) | |
| Stage,[g] n (%) | | | 0.77[b] |
| Stage 1/2 | 14 (58) | 39 (63) | |
| Stage 3/4ab | 10 (42) | 21 (34) | |
| Stage 4/4c | 0 (0) | 1 (1.6) | |
| Unknown | 0 (0) | 1 (1.6) | |

[a]n (%); for three patients from the validation set, each two samples were available.
[b]Pearson's chi-squared test.
[c]High-risk HPV negative.
[d]Median (IQR).
[e]Age missing for one patient.
[f]Wilcoxon rank-sum test.
[g]AJCC TNM8.

## Healthy donor plasma sample collection

Healthy volunteers were recruited in Heidelberg, Germany, and both EDTA and sodium citrate blood plasma samples were collected by medically trained personnel at the German Cancer Research Center. Anonymized blood samples were processed within 1 h after collection by centrifugation at $600 \times g$ for 10 minutes. The plasma phase was separated and centrifuged again at $1,200 \times g$ for 10 minutes. The resulting plasma was split into 1-mL aliquots and stored at −80°C. For each experiment, a minimum of five plasma sample aliquots were pooled and thoroughly mixed before proceeding.

## cfDNA isolation and purification efficiency assessment

Before nucleic acid isolation, plasma samples were equilibrated to room temperature and centrifuged at $3,000 \times g$ for 5 minutes. Automated cfDNA purification was performed using the DNA and Viral NA Large Volume kit for the MagNA Pure 96 (Roche Diagnostics, Mannheim, Germany). cfDNA was purified from EDTA plasma using the cfNA ss 2000 or cfNA ss 4000 protocol, depending on the available plasma volume, and the Viral NA Universal 500 or Viral NA Universal 1,000 protocol for sodium citrate plasma, as

recommended by the manufacturer. The sample volume was assessed based on the sample weight and then adjusted to the necessary input volume for each protocol by adding phosphate-buffered saline (PBS) (e.g., 500 µL plasma plus 1,500 µL of PBS for the cfNA ss 2000 protocol), as recommended by the manufacturer. Purified DNA was eluted in 50 µL elution buffer.

The isolation efficiency of the MagNA Pure 96 protocols was compared to the QIAamp ccfDNA/RNA (QIAamp) kit (QIAGEN, Hilden, Germany) to identify the ideal protocol for cfDNA isolation. Isolation by the QIAamp kit was conducted according to the manufacturer's instructions, and 20 µL of RNase-free water was used in the final elution step. To identify the optimal DNA purification method, HPV16 DNA was spiked into blood plasma from healthy volunteers (spike-in experiments). $10^4$ copies of a 104-bp HPV16 E6 DNA fragment quantified by Agilent Bioanalyzer (Agilent High Sensitivity DNA Kit) were added to pooled plasma samples from healthy volunteers immediately prior to cfDNA isolation. The pooled plasma samples were then thoroughly mixed and split into several aliquots to conduct cfDNA isolation and quantification in multiple replicates across the different kits. For each replicate, the same volume of plasma was used and adjusted with PBS to the minimum input volume of the kit, as described above. As a reference, the original material used for spike-in was separately quantified in each QIAcuity dPCR run. Isolation efficiency was calculated as the number of copies of spike-in DNA detected after isolation divided by the number of copies detected in the original material used for spike-in.

## Testing HPV type specificity on cervical swab samples

Cervical swabs were collected at a colposcopy clinic in Berlin (MVZ) and stored in PreservCyt buffer at 4°C (28). Samples used were HPV negative ($n$ = 21) or positive ($n$ = 20) for HPV16, HPV33 ($n$ = 11), both HPV16 and HPV 33 ($n$ = 2), HPV58 ($n$ = 8), HPV31 ($n$ = 19), HPV35 ($n$ = 2), or HPV52 ($n$ = 9) as determined by multiplex papillomavirus genotyping assay (29). Samples were vortexed vigorously for 20 seconds, and 250 µL was used for DNA purification. PBS was used to adjust the volume to the required input volume of the kit used. DNA was isolated using the MagNA Pure 96 Pathogen Universal 500 protocol, using 50 µL of elution buffer. One microliter of the eluate was used for PCR.

## DNA quantification by dPCR and ddPCR

Isolated DNA was analyzed using both the QX200 ddPCR (BioRad) and the QIAcuity dPCR (QIAGEN). Both digital PCR assays quantified HPV16 E6 and the human reference gene BG using specific primers and probes (duplex assay) (30). On the QIAcuity One, which comes with multiple fluorescence channels to quantify up to five different targets, the assay was complemented by primer-probe sets specific for HPV33 E6 (forward primer: CCAAGCATTGGAGACAACTATAC, probe: TEX-CAACATTGAAC-TACAGTGCGTGGAATG-BHQ2, reverse primer: AATCATATACCTCAGATCGTTGC, reference M12732.1) and HPV16 E7 (forward primer: AAGCAGAACCGGACAGAG, probe: HEX-TGTTGCAAGTGTGACTCTACGCTTCG-BHQ1, reverse primer: TCTACGTGTGTGCTTTGTACG, reference K02718.1 ), thus targeting four different targets (four-plex assay). These primers were designed by identifying the most conserved regions in 29 HPV33 E6 and 638 HPV16 E7 sequences available on the National Center for Biotechnology Information GenBank. Criteria for primer and probe design included suitable melting temperature (ca. 60°C) as calculated by the nearest-neighbor method according to Allawi and SantaLucia (31), a maximum GC content of 70%, and minimal self- and cross-complementarity to prevent the formation of primer dimers. ddPCR on the QX200 system was conducted in 40 cycles of 94°C denaturation for 30 seconds and annealing and elongation at 58°C for 1 minute. dPCR on the QIAcuity One was conducted in 40 cycles of 95°C denaturation for 10 seconds and 60°C annealing and elongation for 30 seconds. For the comparison of the two PCR platforms using the training set, an input of 10 µL of cfDNA was added per reaction, while in the validation set, where dPCR was run on QIAcuity alone, the volume was 20 µL per reaction. As positive controls and for dilution series, plasmids carrying

the full HPV16 and HPV33 genomes and BG were used (32, 33). The positive control contained $10^4$ copies of HPV16, HPV33, and human BG. Dilution series spanned from 1.1 $\times$ $10^5$ to 1.9 copies of those templates in threefold dilution steps for determining the analytical sensitivity of the assay.

## Statistical analysis

Initial analysis of the QIAcuity data was conducted using the QIAcuity Software Suite v.2.2. Results exported from the software were further analyzed in R v.4.3.1 (34) using RStudio (RRID: SCR_000432). Packages from the tidyverse family of R packages (35), epiR (36), and caret (37) were used. In the cervix, HPV infection is common, but extremely low viral loads are most likely clinically irrelevant and may not have been detected by reference tests. Therefore, in cervical swab samples, all samples were only considered HPV positive if a minimum of three partitions was positive for any HPV target. As even in blood plasma samples from patients with HPV-driven malignancies low amounts of HPV DNA are expected, in the training set of plasma samples, any sample with any positive partition was classified as HPV positive. cfDNA samples with only three or fewer "rain" partitions, which are neither clearly positive or negative, were classified as cfDNA negative as these values are most commonly caused by artifacts. Based on the results of the training set, cfDNA samples were only classified as HPV positive when both replicates were HPV cfDNA positive in the validation set of plasma samples to ensure sufficient specificity of the assay. In the assessment of analytical sensitivity, the limit of detection (LoD) was defined as the lowest input copy number that yielded a positive result in both duplicates. Clinical sensitivity was defined as the number of samples from patients with HPV16- or HPV33-driven tumors that tested positive for HPV cfDNA divided by the number of all patients with HPV16- or HPV33-driven tumors. Clinical specificity was defined as the number of samples from HPV16- and HPV33-negative patients which tested negative for HPV cfDNA divided by the total number of HPV16 and HPV33-negative patients. For sensitivity and specificity, the Clopper-Pearson confidence interval (CI) was computed.

## RESULTS

### Comparison of PCR platforms

#### Analytical sensitivity and HPV type specificity

The analytical sensitivity of QX200 ddPCR and QIAcuity dPCR was compared by applying a threefold serial dilution of the corresponding template plasmid DNA in duplicates. Since the QX200 ddPCR assay quantifies BG and HPV16 E6 only, these two targets were used for a direct assay comparison. The QX200 ddPCR had an LoD of 17 copies for both HPV16 E6 and BG (Fig. 1A). The QIAcuity dPCR (Fig. 1A and B) had a lower LoD of six copies for HPV16 E6, HPV16 E7, and BG and a LoD of 17 copies for HPV33 E6.

To assess the specificity of the assay on both PCR platforms, DNA from 92 cervical swab samples with known HPV infection status was tested. Any sample with three or more positive partitions was counted as positive for the given target. In the QX200 ddPCR, HPV16 E6 was 100% specific ($n = 77$ of 77, Fig. 1C). Sensitivity was 95.5% ($n = 21$ of 22 detected), based on one false-negative sample that was defined as HPV16 and HPV33 positive by the reference method and yielded less than three HPV16-positive partitions. In QIAcuity dPCR, HPV16 E6 was 100% type specific and 100% sensitive for both HPV16 single infections ($n = 20/20$) and co-infections with HPV33 ($n = 2$ of 2, Fig. 1C). However, 12 non-HPV16 cases (out of 77) showed one to two partitions positive for HPV16 E6. HPV16 E6 quantification by the two methods was highly correlated (Pearson's $r = 0.97$) and virtually identical (slope of regression model = 0.93).

In QIAcuity dPCR, HPV16 E7 and HPV33 E6 were quantified in the same reaction. HPV16 E7 had a sensitivity of 100% for HPV16 single infections ($n = 20$) and double infections with other HPV types ($n = 2$) (Fig. 1D). HPV33 E6 had a sensitivity of 100% for HPV 33 single infections ($n = 11$) and double infections ($n = 2$). While specificity for HPV16

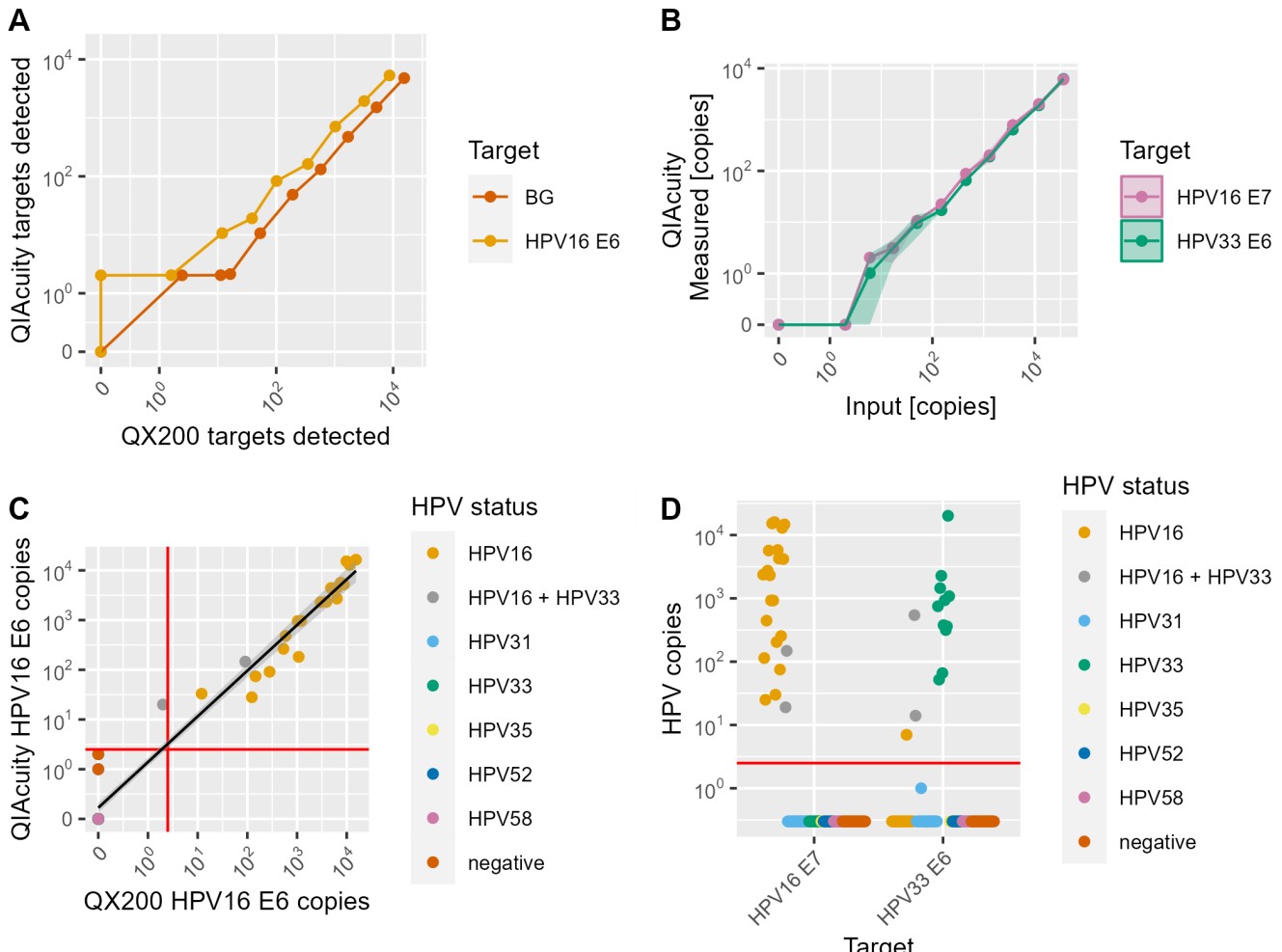

**FIG 1** (A) Quantification of a dilution series of HPV16 E6 and beta-globin (BG) template DNA on a QX200 ddPCR system (*x*-axis) compared to QIAcuity dPCR (*y*-axis). Values shown are the minimum value out of two replicates of each sample. (B) Quantification of a dilution series of HPV16 and HPV33 DNA on QIAcuity dPCR by primers targeting HPV16 E7 and HPV33 E6. Values shown are the minimum of duplicates; the shaded area indicates the standard deviation. (C) Copy numbers of HPV16 E6 detected by QX200 ddPCR (*y*-axis) and QIAcuity dPCR (*x*-axis) in cervical swab samples with known HPV status. The red lines indicate the cutoff of three positive partitions to distinguish positive and negative samples. The black line is a regression line fitted to the data ($y = 0.11 + 0.93x$, $r = 0.97$). Not all colors are visible due to data overlay at zero copies. (D) Copy numbers of HPV16 E7 and HPV33 E6 detected by QIAcuity dPCR in cervix samples with known HPV status. The red line indicates the cutoff of three copies to distinguish positive and negative samples.

E7 was 100%, one sample with an HPV16 infection tested positive for HPV33 but with only seven positive partitions, which is the lowest number detected in any sample. Consequently, the specificity of the HPV33 E6 assay was 98.8% (*n* = 78 of 79). Overall, no HPV DNA was detected in any sample that previously tested negative for HPV DNA or for HPV types other than HPV16 and HPV33.

### cfDNA analysis in a training set of 24 OPC patients

Twenty-four plasma samples from OPC patients (12 HPV16 driven, 2 HPV33 driven, and 10 HPV negative) were purified using the MagNA Pure 96 cfNA ss 2000 protocol and analyzed to compare the QX200 ddPCR targeting HPV16 E6 and BG, and the QIAcuity dPCR targeting HPV16 E6 and BG, and HPV16 E7 and HPV33 E6 in addition. All samples were tested in duplicate.

When tested in ddPCR on the QX200 system, all samples were positive for BG in both replicates with a mean of 6,022 copies/mL detected (range 371–30,678 copies/mL), indicating sufficient DNA amount and quality. In total, 13 out of 24 samples had any

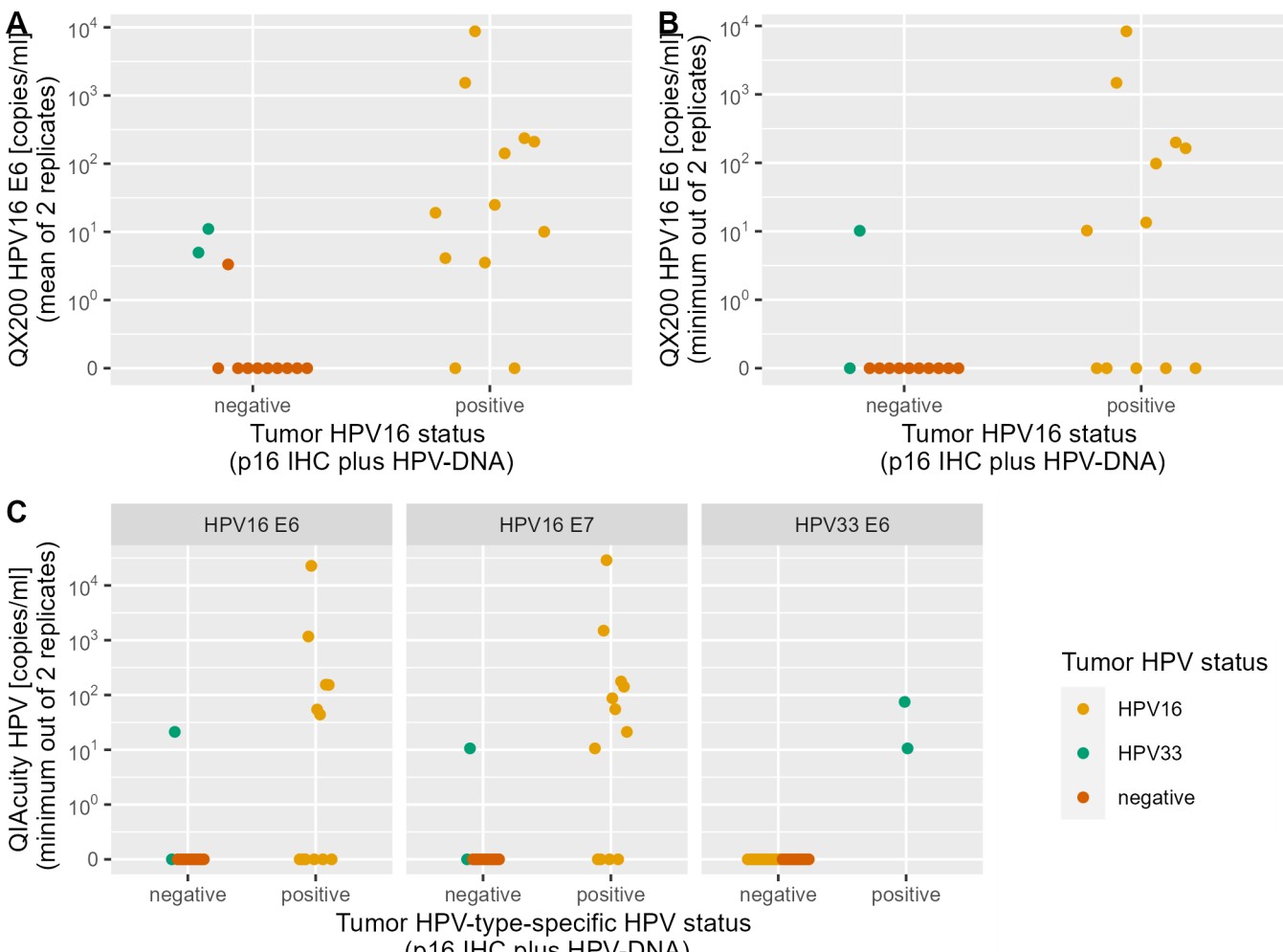

**FIG 2** dPCR results of 24 patients from the training set (10 HPV−, 12 HPV16+, and 2 HPV33+). Samples categorized as positive or negative, depending on the HPV type targeted by each experiment. (A) Copy number of HPV16 E6 cfDNA by QX200 ddPCR, showing the mean value of the replicates. (B) Copy number of HPV16 E6 cfDNA by QX200 ddPCR, showing the minimum value of the replicates. (C) Copy number of HPV16 E6, HPV16 E7, and HPV33 E6 cfDNA by QIAcuity dPCR, showing the minimum value of the replicates.

positive droplets for HPV16 E6 (range 3.33–8,750.0 copies/mL, Fig. 2A). Three of these were false positive with low HPV16 E6 copies (3.33–11.0 copies/mL), yielding a specificity of 75% (9 of 12, 95% CI: 43%–95%). Two of these were positive for HPV33, but not HPV16, in tumor tissue. The assay had a sensitivity of 83% for HPV16-driven OPC cases (10 of 12, 95% CI: 52%–98%). If only samples that were positive for HPV16 in both replicates were counted as HPV16 positive, the sensitivity decreased to 58% (7 of 12, 95% CI: 28%–62%) and specificity increased to 92% (11 of 12, 95% CI: 62%–100%) (Fig. 2B). The lowest concentration that reproducibly tested positive was 11 copies/mL.

When the assay was run with the extended dPCR target panel on the QIAcuity One, a mean concentration of 6,235 copies/mL of BG/mL was detected (201–55,220 copies/mL), indicating sufficient DNA amount and quality in all samples. The assay had an overall sensitivity of 71% (10 of 14, 95% CI: 42%–92%) for HPV16 (E6 and/or E7) and HPV33, and a specificity of 100% (10 of 10, 95% CI: 69%–100%) (Fig. 2C). HPV33 E6 had a sensitivity of 100% (2 of 2, 95% CI: 16%–100%) for HPV33-driven OPC cases and a specificity of 100% (22 of 22, 95% CI: 91%–100%). HPV16 E6 had a sensitivity of 50% (6 of 12, 95% CI: 21%–79%) and HPV16 E7 of 66% (8 of 12, 95% CI: 35%–90%) for HPV16-driven OPC cases. The positive samples for each of the targets contained a mean of 3,482 copies of

HPV16 E6 (range 21.2–22.0, 780 copies/mL), 3,431 copies of HPV16 E7 (range 10.6–28,880 copies/mL) and 42.7 copies of HPV33 E6 (range 10.6–74.8 copies/mL). Six samples were positive for HPV16 E6 and E7, and two additional samples were positive for HPV16 E7 only, indicating a better performance of the HPV16 E7 assay. The quantification of HPV16 E6 and E7 was highly correlated (Pearson's $r > 0.99$, Fig. 3A). Cohen's kappa was 0.476 for HPV16 E6, 0.645 for HPV16 E7, and 1.0 for HPV33 E6, indicating moderate to perfect agreement with tumor HPV status. There was no correlation of HPV16 E7 copy numbers with BG copy numbers (Pearson's $r = 0.01$, Fig. 3B). There was a weak correlation between the number of BG copies detected and the plasma volume (Pearson's $r = 0.221$, Fig. 3C).

Both HPV16 assays had a specificity of 92% due to one false-positive result. One plasma sample from an HPV33-driven OPC case reproducibly tested positive for HPV16 E6 (two of two replicates on QIAcuity, two of two replicates on QX200) and HPV16 E7 (two of two replicates on QIAcuity), in addition to HPV33 E6 (two of two replicates), likely indicating that an HPV16 infection was missed in the tumor tissue, or contamination occurred.

## Optimization of cfDNA purification

In order to optimally detect low concentrations of cfDNA in small volumes of less than 1 mL of blood plasma, the performance of different nucleic acid purification methods was analyzed. The automated cfNA ss 2000 protocol for the MagNA Pure 96 and the manual QIAamp ccfDNA/RNA kit were chosen as they represent two common technologies used for DNA purification: column-based methods where DNA binds to a silica membrane (QIAamp ccfDNA/RNA) or silica-coated magnetic beads (MagNA Pure 96).

To ensure comparability, one plasma pool from healthy volunteers with spiked-in HPV16 plasmid DNA was split into multiple aliquots and tested on both platforms. Based on the quantification of spiked-in HPV16 DNA by QIAcuity dPCR, mean isolation efficiency was higher using the cfNA protocol for the MagNA Pure 96 cfNA ss 2000 (51%) compared to the QIAamp ccfDNA/RNA kit (23%) (Fig. 4A). Similarly, a higher number of BG copies from the native cfDNA were detected in the MagNA Pure 96

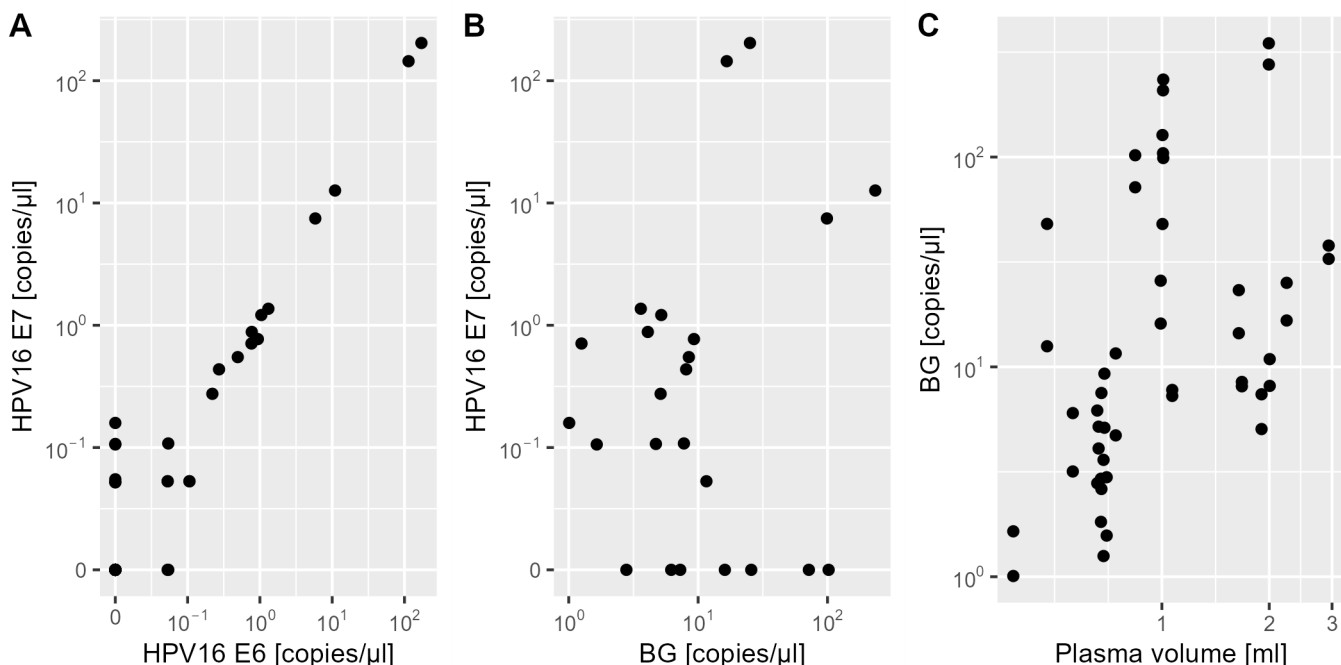

FIG 3 Number of copies of different target DNA detected in plasma samples from 24 OPC patients by QIAcuity dPCR. Each point represents one of two replicates from a sample. (A) Comparison of HPV16 E6 and HPV16 E7 copies detected in each sample. (B) Comparison of copy number of HPV16 E7 and BG copies in each sample. (C) Comparison of number of copies of BG detected and the input blood plasma volume.

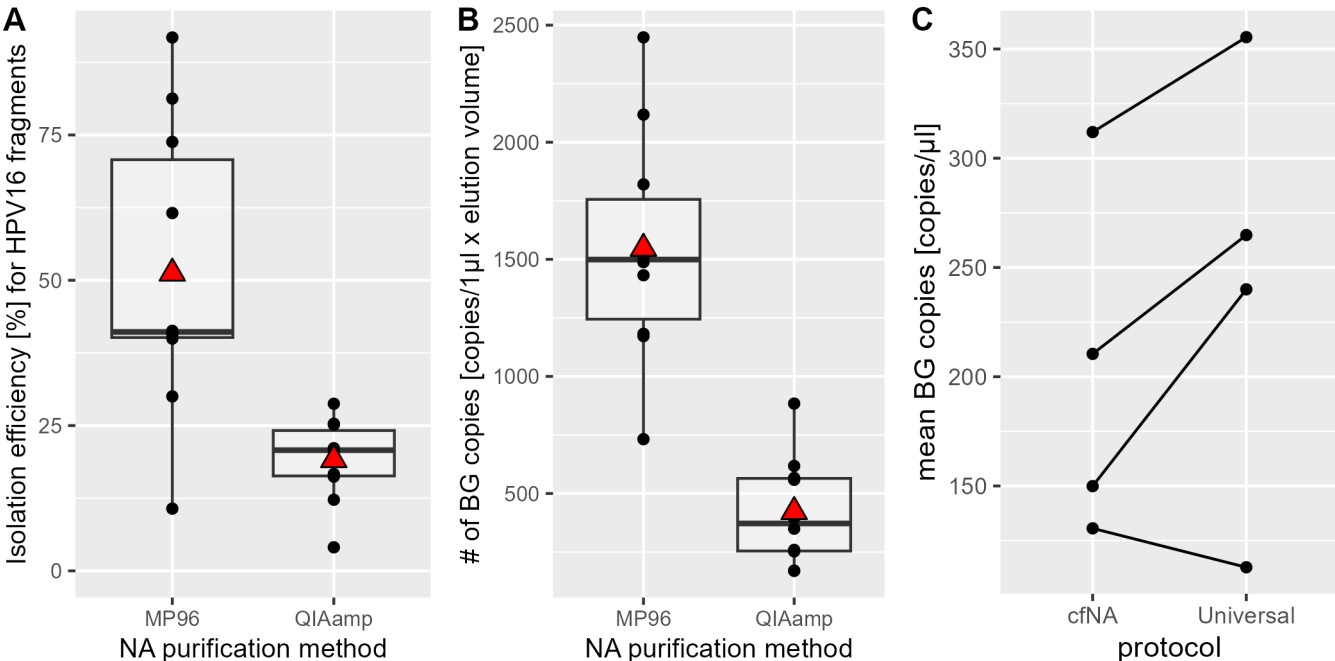

**FIG 4** Performance of nucleic acid isolation from blood plasma. (A) Recovery rate of a 104-bp HPV16 fragment from EDTA plasma by MagNA Pure 96 (MP96) cfNA ss 2000 protocol or QIAamp ccfDNA/RNA kit (QIAamp). The red triangle indicates the position of the group mean. (B) Number of copies of native BG cfDNA isolated from aliquots of one EDTA plasma pool by MP96 and QIAamp. (C) Comparison of MagNA Pure 96 cfNA ss 2000 protocol and the Viral NA Universal 500 protocol for sodium citrate plasma.

cfNA ss 2000 extract (median 1,499 copies/mL) compared to the QIAamp ccfDNA/RNA extract (median 372 copies/mL) (Fig. 4B). When using the MagNA Pure 96 cfNA ss 2000 protocol for isolating HPV DNA spiked into healthy donor EDTA plasma, as few as 10 molecules of HPV DNA were detected in eight of eight replicates (100% sensitivity) in the subsequent QIAcuity dPCR, while with five molecules, the sensitivity was 75% (six of eight replicates). When cfDNA was isolated from sodium citrate plasma samples from healthy volunteers, the MagNA Pure 96 cfNA ss 2000 protocol yielded fewer copies of BG per milliliter of plasma in three out of four samples than the Viral NA Universal 500 protocol, despite using the same input volume in both protocols (Fig. 4C). Consequently, the latter protocol was used for the sodium citrate plasma samples in the validation experiment.

**Effect of sample volume on quantification**

To gain insight about the effect of the sample volume on sensitivity, aliquots of 500 and 1,000 µL were analyzed from a subset of 30 plasma samples from the validation set of OPC patients (15 HPV16 positive, 2 HPV58 positive, and 13 HPV negative), which had a minimum of 1.5 mL of plasma. The samples were purified using MagNA Pure 96 Viral NA Universal 500 or 1,000 protocol and analyzed in duplicate by QIAcuity dPCR.

All samples in both groups (500 µL vs 1,000 µL) had sufficient human BG DNA. When 1,000 µL of plasma was used for cfDNA isolation, the HPV16 E7 dPCR had a sensitivity of 80% for HPV16-driven tumors ($n$ = 12 of 15), while HPV16 E6 had a sensitivity of 73% ($n$ = 11 of 15) (Fig. 5). When 500 µL of sample was analyzed, the assay had a sensitivity of 73% ($n$ = 11 of 15) for both HPV16 E6 and E7. In total, only one HPV-driven OPC case was not detected due to decreased input plasma volume. Overall, the quantification of cfDNA copies per milliliter was very consistent between the two sample volumes for HPV16 E6 and E7 ($r \geq 0.99$) and was comparable for human BG ($r = 0.86$) (Fig. 5). The cfDNA isolation efficiency did not vary with sample volume, as indicated by a slope of approximately 1

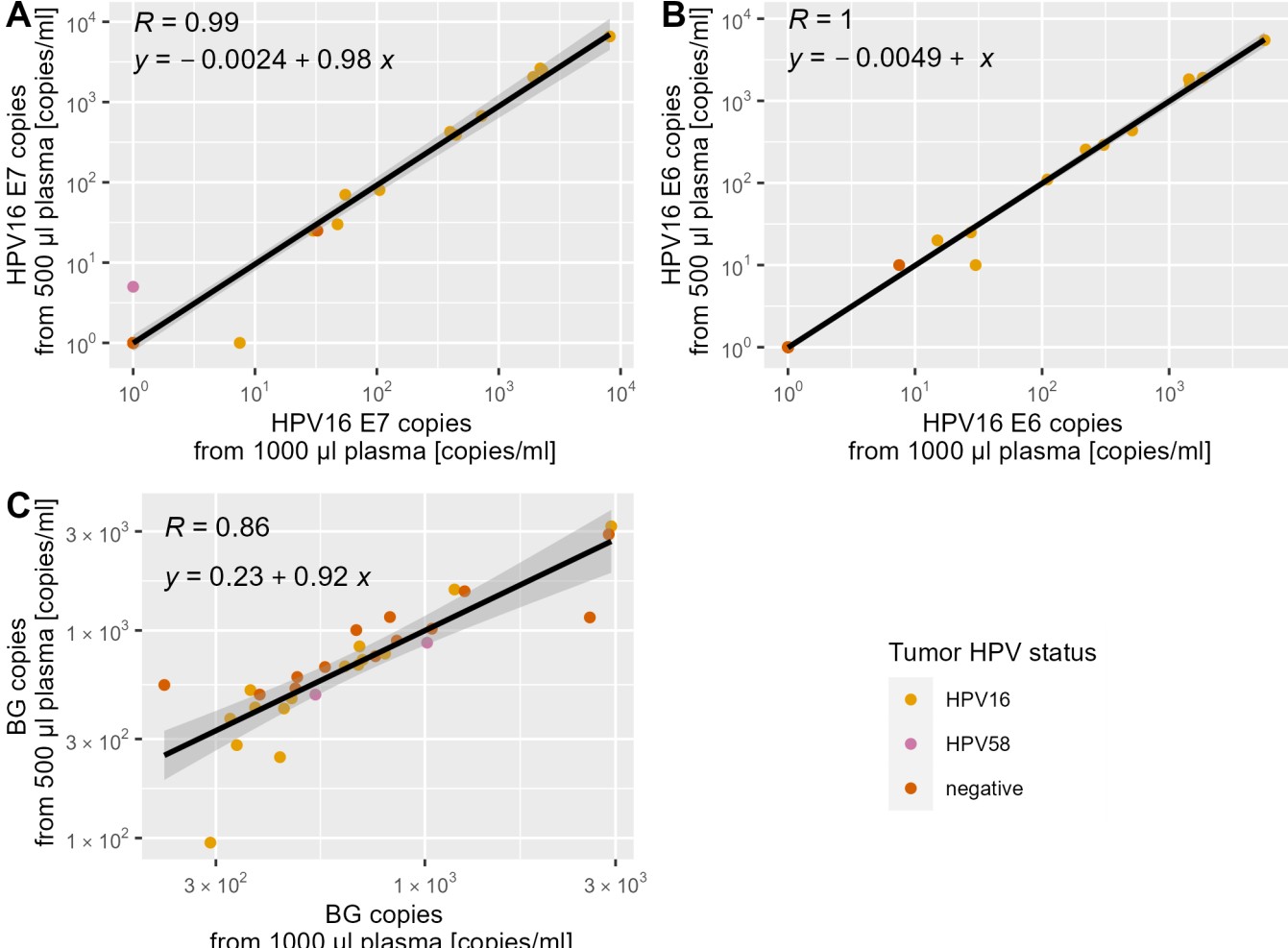

**FIG 5** QIAcuity dPCR result of 30 OPC patient plasma samples (13 negative, 2 HPV58 positive, and 15 HPV16 positive), comparing the concentration of each target (in copies/mL plasma) between cfDNA isolated from 500 µL (*y*-axis) and 1,000 µL (*x*-axis) of plasma. (A) HPV16 E7, (B) HPV16 E6, and (C) BG.

between copy number per milliliter detected in low- and high-volume samples for all three targets (E6, E7, and BG, Fig. 5).

There were two false-positive samples, leading to a specificity of the assay of 87% in this subset of samples ($n$ = 13 of 15). In one HPV58-positive OPC plasma sample, very low amounts (2.5 copies/mL) of HPV16 E7-cfDNA only were detected in the 500-µL extract only. The other false-positive plasma sample was from an OPC case where the corresponding tumor tissue was HPV DNA negative but p16 positive. This sample tested concordantly positive for both HPV16 E6 and E7 cfDNA in both plasma aliquots of 500 and 1,000 µL, likely indicating a missed HPV16 infection in the tumor tissue.

### Validation of final protocol in plasma samples from OPC patients

For the final assessment of clinical sensitivity and specificity, a total of 65 plasma samples from OPC patients (34 HPV16 positive, 1 HPV33 positive, 2 HPV58 positive, and 28 HPV negative) were analyzed using the results from the 1,000-µL plasma volume where available. In the following analysis, HPV58-driven OPC cases were considered HPV negative as the assay was not targeted to detect this HPV type. cfDNA was purified from plasma samples by MagNA Pure 96 Viral NA Universal protocol and applied in duplicate to QIAcuity dPCR. Only samples that tested reproducibly positive for a given target were considered positive for this target, based on results from the training set. Sample

**TABLE 2** Performance of HPV QIAcuity dPCR of plasma cfDNA in comparison to tumor HPV status

| cfDNA[a] | Tumor HPV status (HPV DNA + p16-IHC) | |
|---|---|---|
| | HPV16+ | HPV16− |
| HPV16 E7+ | 27 | 2 |
| HPV16 E7− | 7 | 29 |
| | HPV16+ | HPV16− |
| HPV16 E6+ | 22 | 2 |
| HPV16 E6− | 12 | 29 |
| | HPV33+ | HPV33− |
| HPV33 E6+ | 1 | 0 |
| HPV33 E6− | 0 | 64 |
| cfDNA[a] | HPV16/33+ | HPV16/33− |
| HPV+[b] | 28 | 1 |
| HPV−[b] | 7 | 29 |
| Sensitivity | 0.800 (95% CI: 0.63–0.92) | |
| Specificity | 0.967 (95% CI: 0.83–1.00) | |
| Kappa | 0.756 (95% CI: 0.68–0.84) | |

[a]HPV detected in plasma by QIAcuity dPCR.
[b]HPV+/HPV− summarizes the results of HPV16 E7, HPV16 E6, and HPV33 E6 dPCR; the overall result was positive for HPV if any of the three HPV targets was detected in both duplicates for a sample.

volumes analyzed for each sample ranged from 370 to 1,000 µL (mean of 824 µL). The samples contained a median of 684 copies/mL of BG (range 223–80,982 copies/mL).

Samples diagnosed as HPV16 or HPV33 driven by pathology were classified correctly with a sensitivity of 80% (28 of 35, 95% CI: 63%–92%, Table 2) and with a specificity of 96.7% (29 of 30, 95% CI: 83%–100%). The test had a Cohen's kappa of 0.756, indicating substantial agreement with the tumor HPV status. In samples with 350–500 µL of plasma available, the sensitivity of the assay was 73.3% (11 of 15, 95% CI: 45%–2%), while in samples with 750–1,000 µL of plasma, the sensitivity was 85% (17 of 20, 95% CI: 62%–97%). In plasma samples from HPV-driven tumors with nodal stage 0, the sensitivity was 75% (9/12, 95% CI: 43 - 95%); in samples with nodal stage 1, the sensitivity was 82% (14 of 17, 95% CI: 57%–96%); and in samples with nodal stage 2, the sensitivity was 75% (3 of 4, 95% CI: 19%–99%). A moderate correlation between N stage and HPV16 E7 copy numbers detected in plasma samples from HPV16-driven OPCs was observed ($r = 0.272$), while there was a weak correlation with T stage ($r = 0.128$). There was no correlation between BG copy numbers detected and the sensitivity of the assay, but five out of five cases with more than 10,000 copies of beta-globin were HPV driven. The age of the sample had no effect on the number of copies of BG detected in the plasma samples ($r = -0.076$). Sensitivity of HPV detection was 86% (6 of 7, 95% CI: 42%–100%) in samples less than 3 years old, 67% (10 of 15, 95% CI: 38%–88%) in samples 3–6 years old, and 92% (12 of 13, 95% CI: 64%–100%) in samples older than 6 years. As in all these groups similar sample volumes were used (mean plasma volumes of 786, 767, and 742 µL, respectively), sample storage for up to 9 years did not have a measurable impact on the sensitivity of HPV detection in these samples.

## DISCUSSION

Early detection of OPC may hold the potential to improve a patient's quality of life due to early therapeutic intervention requiring less intensive therapy. For this purpose, novel diagnostic molecular biomarkers are needed using minimally invasive protocols. HPV cfDNA is a promising candidate that is successfully being used in therapy surveillance of HPV-driven OPC already. To evaluate this biomarker for early diagnosis, a highly sensitive and specific assay is needed that can also be successfully applied to archived material, where usually only small amounts of plasma are available. Here, we describe a digital PCR assay targeting HPV16 and HPV33.

To optimize the assay at every step from DNA isolation from plasma to DNA detection and quantification, multiple comparisons of methods were conducted. Due to the superior cfDNA isolation efficiency of the MagNA Pure 96 over the QIAamp kit, the MagNA Pure 96 was used in the final validation set of samples. While the QX200 and QIAcuity showed comparable analytical and clinical sensitivity, the QIAcuity offers higher multiplexing capacity, which enables the targeting of multiple HPV types at once. In addition, the effect of sample volume on sensitivity was evaluated. Our results suggest that it is feasible to use lower input volumes than seen in most studies, as using only up to 1 mL of plasma in the validation experiment yielded a sensitivity of 80%, which is comparable to other studies which achieved 70%–96% sensitivity working with 2–4 mL of plasma per sample (13, 19, 20, 38). Since even in samples with 500 µL or less plasma available the sensitivity was still 73.3%, the assay is suitable for use in archived samples from large collections in biobanks, where usually only small volumes are available.

The increase in sensitivity from the training set (71%) to the validation set (80%) could be explained by an increased PCR input volume from 10 to 20 µL due to increased sample volume requirements in the training set for the comparison of two different dPCR methods and the use of an optimized cfDNA purification method in the validation set. The unexpectedly low cfDNA yield in samples with high input volumes from the training set prompted a more in-depth comparison of different cfDNA isolation protocols for sodium citrate plasma. This led to the conclusion that the Viral NA Universal protocols give a higher yield in these samples than the cfDNA protocols, so the protocol used for cfDNA isolation was changed in the validation set.

The specificity of the QIAcuity dPCR in the final validation set was 96.6%. The one reproducibly false-positive HPV16 E6 and E7 cfDNA plasma sample by dPCR was from a patient with a tumor that was tested HPV DNA negative and p16 positive by a pathologist. The same plasma sample has previously tested positive for antibodies against multiple HPV16 antigens using multiplex serology, which is an assay that has a very high specificity (39, 40). Therefore, it seems likely that the HPV DNA in the tumor tissue has been missed by the reference assay, and it could be reasonably assumed that the tumor was actually HPV driven, which would set the actual specificity of the assay to 100%, even though cross-contamination cannot be completely ruled out.

In the training set and the experiment for the direct comparison of cfDNA isolated from different volumes, three additional false-positive cases emerged. In contrast to the HPV DNA result of corresponding tumor tissue samples, an additional HPV16 DNA was detected in three plasma samples from HPV33- or HPV58-driven OPC patients. This could be explained by either a lack of type specificity of the assay when applied to cfDNA, which seems unlikely as there was no cross-reactivity in cervical swab samples positive for any of these HPV types, or previously undetected co-infections in these cases due to sensitivity limitations of the reference test. In cervical swabs within this study, there were multiple cases of co-infection with these HPV types, which have also been frequently observed in other studies (41), although whether these findings are applicable to the oropharynx requires further investigation.

Although HPV16 is the most common HPV type among HPV-driven OPC, one limitation of the assay is the small number of HPV types covered. In our study, HPV58 was present almost as frequently as HPV33. However, only HPV33 was included in the assay due to the limited multiplexing spectrum of the dPCR. Since all HPV16 E6 positive samples were also HPV16 E7 positive, in future studies, HPV16 E6 could be replaced by primers and probes detecting other HPV types.

Since therapy surveillance is a well-established use for HPV cfDNA detection, a more interesting question to answer would be the kinetics of HPV cfDNA before diagnosis of HPV-driven cancer. The possibility of using lower input volumes may open up new sample sources that would not be able to provide the volumes usually used for cfDNA isolation. The assay presented in this study is sensitive and specific even for these smaller input volumes and thus could help in exploring the utility of HPV cfDNA in early

detection of HPV-driven cancers, which could be feasible due to only requiring minimally invasive sampling, especially in combination with HPV early antigen serology.

## Conclusions

We established a dPCR assay and workflow detecting HPV16 E6, HPV16 E7, and HPV33 E6 in archived, low-volume plasma samples. This workflow was shown to detect sensitively and specifically HPV16 and HPV33. Follow-up studies in larger cohorts are required to explore the use of HPV cfDNA in early detection of HPV-driven OPC.

## ACKNOWLEDGMENTS

This study was funded by intramural German Cancer Research Center funding.

## AUTHOR AFFILIATIONS

[1]Infections and Cancer Epidemiology, German Cancer Research Center (DKFZ), Heidelberg, Germany

[2]Department of Otorhinolaryngology and Head and Neck Surgery, Head and Neck Cancer Center of the Comprehensive Cancer Center Ulm, University Medical Center Ulm, Ulm, Germany

[3]Department of Gynecology, HPV Research Laboratory, Charité-Universitätsmedizin Berlin, Corporate Member of Freie Universität Berlin and Humboldt-Universität zu Berlin, Berlin, Germany

[4]Department of Otorhinolaryngology, Head and Neck Surgery, Medical Faculty, University of Cologne, Cologne, Germany

[5]Department of Internal Medicine, Faculty of Medicine, Center for Integrated Oncology Aachen Bonn Cologne Duesseldorf, University Hospital Cologne, University of Cologne, Cologne, Germany

[6]Department of Radiobiology and Radiation Oncology, University Medical Center Hamburg-Eppendorf, Hamburg, Germany

[7]Department of Otolaryngology and Head and Neck Surgery, University Medical Center Hamburg-Eppendorf, Hamburg, Germany

[8]Department of Otolaryngology-Head and Neck Surgery, Harvard Medical School, Boston, Massachusetts, USA

[9]Mass Eye and Ear, Boston, Massachusetts, USA

## AUTHOR ORCIDs

Daniel L. Faden http://orcid.org/0000-0001-5284-7762
Daniela Höfler http://orcid.org/0000-0001-9458-5190

## FUNDING

| Funder | Grant(s) | Author(s) |
|---|---|---|
| Deutsches Krebsforschungszentrum (DKFZ) | | Fabian Rosing |
| | | Lea Schroeder |
| | | Tim Waterboer |
| | | Daniela Höfler |

## DATA AVAILABILITY

The data generated in this study are available upon request from the corresponding author.

## ETHICS APPROVAL

The experiments in this publication have been approved by ethics committees of both Heidelberg University (Ethics Commission, ethical approval S-031/2022) and Ulm University (Ethics Commission, ethical approval 90/15). All participants provided written informed consent.

## ADDITIONAL FILES

The following material is available online.

### Open Peer Review

**PEER REVIEW HISTORY (review-history.pdf).** An accounting of the reviewer comments and feedback.

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
