## [Reviewer comments · Microbiology Spectrum]

Microbiology Spectrum

Quantification of Human Papillomavirus cell-free DNA from low volume blood plasma samples by digital PCR

Fabian Rosing, Matthias Meier, Lea Schroeder, Simon Laban, Thomas Hoffmann, Andreas Kaufmann, Oliver Siefer, Nora Wuerdemann, Jens Klussmann, Thorsten Rieckmann, Yvonne Alt, Daniel Faden, Tim Waterboer, and Daniela Höfler

Corresponding Author(s): Daniela Höfler, Deutsches Krebsforschungszentrum

Review Timeline:

Submission Date:	January 23, 2024
Editorial Decision:	February 19, 2024
Revision Received:	April 23, 2024
Accepted:	April 23, 2024

Editor: Peter Pelka

Reviewer(s): The reviewers have opted to remain anonymous.

Transaction Report:

DOI: <https://doi.org/10.1128/spectrum.00024-24>

Re: Spectrum00024-24 (Quantification of Human Papillomavirus cell-free DNA from low volume blood plasma samples by digital PCR)

Dear Dr. Daniela Höfler:

Thank you for the privilege of reviewing your work. Below you will find my comments, instructions from the Spectrum editorial office, and the reviewer comments.

I have now received reviews of experts in the field, which are appended below. Reviewer #1 had two additional suggestions that should be addressed before publication. Thank you for submitting to Spectrum.

Revision Guidelines

Sincerely,
Peter Pelka
Editor
Microbiology Spectrum

Reviewer #1 (Comments for the Author):

The authors have established a dPCR assay and workflow for detecting HPV16 and HPV33 E6 in low volume plasma samples from OPC patients. This workflow was shown to detect sensitively and specifically HPV16 and HPV33. The article is very well written, and of a great interest for the scientific community as there is an urgent need to develop and

validate body-fluids based screening methods for the early detection of HPV-driven OPCs, and screening.

Minor revisions:

1- The authors have used low volume plasma samples with defined HPV status stored for 2-9 years. Did the sensitivity vary with the storage time? Would it be possible to perform this correlation?

2- Most of the HPV genotyping assays are based on L1 gene. Why did the authors target E6 or E7 genes for HPV16 and HPV33 detection?

Reviewer #2 (Comments for the Author):

Some specific comments:

A section in the discussion that talks about other potential methods for cfDNA purification; along with the merits and drawbacks would be useful - especially as it relates to the 2 methods tested in this study. Why did the authors pick these 2 methods to test and not others? This could be described better to give the readers more insight into the choices made for this study.

The volume of serum used to extract the cfDNA seems to be critical. Perhaps alluding to the importance of conducting an experiment where cfDNA isolation efficiency as a function of serum volume (spiked healthy donor) can be assessed to determine optimal volumes would be necessary here.

Overall the experiments are described well, and the conclusions are justified by the results.

To
Microbiology Spectrum

Response to Reviewers

Heidelberg, 02/03/2024

Dear Reviewers,

firstly, I would like to thank you for taking the time to review this manuscript, your feedback is much appreciated. I have carefully read your comments and have addressed any issues you have pointed out. Below you can find my point-by-point response to your comments.

Reviewer #1:

“The authors have used low volume plasma samples with defined HPV status stored for 2-9 years. Did the sensitivity vary with the storage time? Would it be possible to perform this correlation?”

This is an interesting question which can be answered based on our data, so information regarding this question was added in section 3.4. In short, sensitivity does not follow a clear trend with age of the sample, so it does not seem to be influenced by storage times up to 9 years.

Reviewer #1:

“Most of the HPV genotyping assays are based on L1 gene. Why did the authors target E6 or E7 genes for HPV16 and HPV33 detection?”

It is true that many HPV assays, especially those used for cervical cancer screening, are based on L1. Nevertheless, we compared the performance of three different L1 and E6/E7 based (q)PCR methods in cervical (pre-) cancerous swabs and did not find significant differences in detecting high grade lesions (unpublished data). In addition, other publications have found the E6 and E7 region of the HPV genome are highly conserved regions within the HPV genome and are often amplified in cancer ([10.1101/gr.164806.113](https://doi.org/10.1101/gr.164806.113)), which reduces the risk of a false negative result by primers and probes not binding due to sequence variations in the primer target regions or lack of target sequence. As E6 and E7 are necessary oncogenes that drive growth of HPV-driven carcinomas, these genes would very rarely be deleted in HPV-driven carcinoma cells and consequently not missed by a PCR targeting these genes. We now addressed your question in the introduction of the manuscript.

Reviewer #2:

“A section in the discussion that talks about other potential methods for cfDNA purification; along with the merits and drawbacks would be useful - especially

Foundation under Public Law

Management Board
Prof. Dr. med. Dr. h. c. Michael Baumann
Ursula Weyrich

Deutsche Bank Heidelberg
IBAN: DE09 6727 0003 0015 7008 00
BIC (SWIFT): DEUT DES M672

Deutsche Bundesbank Karlsruhe
IBAN: DE39 6600 0000 0067 0019 02
BIC (SWIFT): MARK DEF 1660

as it relates to the 2 methods tested in this study. Why did the authors pick these 2 methods to test and not others? This could be described better to give the readers more insight into the choices made for this study.”

This is an important point that is now addressed in the results section 3.2 of the manuscript. The two methods chosen were chosen to represent the main two technologies used for DNA purification – column-based methods (QIAmp kit) and magnetic bead-based methods (MagNA Pure).

Reviewer #2:

“The volume of serum used to extract the cfDNA seems to be critical. Perhaps alluding to the importance of conducting an experiment where cfDNA isolation efficiency as a function of serum volume (spiked healthy donor) can be assessed to determine optimal volumes would be necessary here.”

The volume of blood plasma used is indeed highly important for the sensitivity of cfDNA analysis. Therefore, we included the comparison of sensitivity using different sample volumes in section 3.3 of the manuscript. As demonstrating the performance of DNA purification on clinical samples is closer to the intended application of the assay than using spike-in DNA we had decided not to include data about the effect of sample volume on isolation efficiency of spike-in DNA. In the data included in the manuscript it is shown that isolation efficiency is virtually identical between the different volumes used as the number of HPV copies detected per volume was identical (see fig. 5). To emphasize this the result section 3.3 of the manuscript has been edited.

We hope that the edits made to the manuscript are sufficient to qualify it for publication.

Sincerely yours,

Fabian Rosing

Re: Spectrum00024-24R1 (Quantification of Human Papillomavirus cell-free DNA from low volume blood plasma samples by digital PCR)

Dear Dr. Daniela Höfler:

Your manuscript has been accepted, and I am forwarding it to the ASM production staff for publication. Your paper will first be checked to make sure all elements meet the technical requirements. ASM staff will contact you if anything needs to be revised before copyediting and production can begin. Otherwise, you will be notified when your proofs are ready to be viewed.

Sincerely,
Peter Pelka
Editor
Microbiology Spectrum